# Indoor NLOS Positioning System Based on Enhanced CSI Feature with Intrusion Adaptability

**DOI:** 10.3390/s20041211

**Published:** 2020-02-22

**Authors:** Ke Han, Lingjie Shi, Zhongliang Deng, Xiao Fu, Yun Liu

**Affiliations:** School of Electronic Engineering, Beijing University of Posts and Telecommunications, No.10 XiTuCheng Road, Beijing 100876, China; shilj@bupt.edu.cn (L.S.); dengzhl@bupt.edu.cn (Z.D.); Xiaofu@bupt.edu.cn (X.F.); bupt2013211026ly@bupt.edu.cn (Y.L.)

**Keywords:** channel state information, indoor positioning, intrusion detection, non-line-of-sight

## Abstract

With the wide deployment of commercial WiFi devices, the fine-grained channel state information (CSI) has received widespread attention with broad application domain including indoor localization and intrusion detection. From the perspective of practicality, dynamic intrusion may be confused under non-line-of-sight (NLOS) conditions and the continuous operation of passive positioning system will bring much unnecessary computation. In this paper, we propose an enhanced CSI-based indoor positioning system with pre-intrusion detection suitable for NLOS scenarios (C-InP). It mainly consists of two modules: intrusion detection and positioning estimation. The introduction of detection module is a prerequisite for positioning module. In order to improve the discrimination of features under NLOS conditions, we propose a modified calibration method for phase transformation while the amplitude outliers are filtered by the variance distribution with the median sequence. In addition, binary and improved multiple support vector classification (SVC) models are established to realize NLOS intrusion detection and high-discrimination fingerprint localization, respectively. Comprehensive experimental verification is carried out in typical indoor scenarios. Experimental results show that C-InP outperforms the existing system in NLOS environments, where the mean distance error (MDE) reached 0.49 m in the integrated room and 0.81 m in the complex garage, respectively.

## 1. Introduction

With the increasing development of ubiquitous location-based services (LBS), the popularization of intelligent industries accelerates the process of smart application in indoor scenarios [1,2,3]. Considering the huge application prospects for the fact that human activities mainly occur in indoor environments [4], the practical indoor location-based services has currently attracted a great deal of research enthusiasm [5,6]. Limited by incomplete coverage of satellite signals, the performance of Global Positioning System (GPS) that widely used outdoors cannot be well performed indoors, thus diverse indoor positioning technologies have been proposed in the past two decades. The existing techniques can be roughly divided into these categories—vision-based [6,7], sensor-based [2,8] and wireless-based including Wi-Fi [9,10], Bluetooth [11,12], acoustic signals [13], ultra-wideband (UWB) [14,15] and so forth. Among the above technologies employed for indoor localization system, WiFi-based positioning method has attracted increasing attention as a promising technology due to its convenience of flexible deployment, low configuration cost and high availability [10]. Nevertheless, the current WiFi-based passive positioning methods mainly focus on line-of-sight (LOS) scenes and only target the direct propagation path without considering the reflection or transmission.

The accuracy of WiFi-based localization is severely interfered by many factors in typical indoor environments—signal power attenuation, dense multi-path effects, shadow fading, as well as transmission delay, and so forth [16,17]. The widely researched received signal strength indicator (RSSI)-based method is beyond reach of precise characterization of Wi-Fi features due to its coarse-grained characteristics [18], and then the mapping relationship between RSSI and corresponding separate locations cannot be accurately established. The availability of the fine-grained physical layer CSI on commercial Wi-Fi devices has stimulated increasing research enthusiasm [9,19]. CSI depicts frequency diversity [20] at the OFDM subcarrier level and provides detailed CSI fine-grained PHY layer information. Nevertheless, Wi-Fi equipment can be deployed flexibly but limited by hardware defects, the achievement of high-precision positioning performance is still a big challenge.

From methodology perspective, the current WiFi-based indoor positioning technology is mainly divided into two types—mathematical model-based approaches [18] and fingerprint-based methods [9,21,22,23]. The former take advantage of statistical models to achieve localization by establishing a mathematical relationship between the wireless signals and the anchor points, while the other utilize unique pattern differences between respective indoor locations. Wu et al. [18] firstly proposed an indoor propagation model called FILA that the position is obtained according to the geometric relationship analysis, but the drawbacks of rich multipath interference limit the model-based approach to the ideal line-of-sight (LOS) cases only. FIFS system [21] was further proposed using CSI amplitude feature as a fingerprint database and realizing matching progress by exploiting relevant feature differences of the received signals. In order to better characterize the influence of the target on the wireless signal and extract the appropriate statistical features, Wang et al. [23] proposed PhaseFi to transform phase variances for feature preprocessing. Nevertheless, from the perspective of practicality, the passive positioning system needs continuous operation to achieve detection effect, but its high computation cannot be ignored.

High-precision detection is a prerequisite for wireless signal perception [24,25,26]. For instance, behavior recognition [27] and LoS identification [28] require a relatively static environment for the next step of judgment. In order to avoid the interference of dynamic personnel movement on the locating accuracy during the passive positioning process, it is necessary to sense human movement before starting positioning progress, so as to minimize the influence of dynamic human interference on positioning accuracy. These works mostly focus on the purpose of recognition or localization itself while resigning the existence detection of moving human to simple threshold method to achieve simple binary classification, which always result in a large false alarm rate (FAR) in complex NLOS scenarios due to the sensitivity of CSI features. The existing CSI behavior recognition have already made great progress [29,30,31], which mainly focuses on the optimization of signal amplitude and phase characteristics. Wu et al. [25] proposed DeMan, a non-invasive human dynamic and static detection method by taking advantage of both amplitude and phase. In particular, a breathing detection is considered as an intrinsic indicator of static human presence. Domenico et al. [32] explored training options on Doppler spectrum-based approach for motion and presence detection and further conclude that reduced training might not necessarily affect detection performance. They also calculated the CSI subcarrier frequency offset to obtain the average spectrum, and the through-wall personnel can be detected as well [33].

To address these challenges, we propose an enhanced CSI-based indoor positioning system with pre-intrusion detection suitable for NLOS scenarios (C-InP). It mainly focuses on three issues—the processing of CSI anomalies under NLOS conditions, high-precision sequential detection in complex scenarios, and the actual operational complexity. This system consists of two modules—intrusion detection and positioning estimation. Firstly, intrusion detection module is designed to alleviate system calculations, which is considered a prerequisite for positioning, thereby triggering positioning module. Secondly, in order to improve the discrimination of features in NLOS scenes, we propose a modified least square (LS) calibration method for phase transformation while the amplitude outliers are filtered by the variance distribution with the median sequence for feature preprocessing. In detection module, we observe dynamic CSI changes between adjacent time windows to determine dynamic intrusion. In positioning module, we take advantage of the mapping relationship between the processed CSI features and the corresponding location to establish high-discrimination classification model inspired by the support vector machine. In addition, we adopt SVD to realize dimensionality reduction for the improved classification method. Our goal in this work is to make use of Wi-Fi infrastructure to enable low-cost deployment, and the integrated utilization of intrusion and localization technologies to break through the limitations of indoor application scenarios.

In summary, our main contributions are as follows:In order to comprehensively consider the practicability and positioning accuracy, we increase discrimination of both amplitude and phase respectively by filtering the outliers by variance distribution and modified LS calibration method for the linear transformation, simplifying the complex process of signal feature optimization.A unified CSI-based framework for indoor positioning is designed for practical scenarios, which alleviates the actual calculation of the system and improves the positioning accuracy under NLOS conditions relatively.The B-SVC method in C-InP is utilized for confused intrusion detection under NLOS conditions and an improved multi-class fingerprint positioning method is proposed to realize low-complexity for indoor localization.Comprehensive experimental verification is carried out in typical indoor scenarios. Experimental results show that C-InP outperforms the existing system in NLOS environments and effectively reduces the operation time of the system and improves the real-time response.

The rest of this paper is organized as follows. Section 2 briefly present relevant preliminaries and challenges. In Section 3, we describe the architecture overview of the proposed system, details of feature processing of observed eigenvalues and localization algorithm. Section 4 presents the experiment implementation and evaluation. Finally, we draw the conclusions in Section 5.

## 2. Preliminaries and Challenges

### 2.1. CSI Introduction

Compared with RSS in MAC layer, CSI describes the time and spectrum structure of wireless link in terms of amplitude and phase. Specifically, from the frequency domain, CSI is physical layer information with fine-grained attribute values describing the amplitude and phase of each subcarrier. CSI represents the channel characteristics of the communication link, and describes the attenuation of the signal during transmission of the transmitter and receiver in the time domain, including time delay, signal scattering, multipath distortion, and distance attenuation, and so forth.

CSI conforms to the IEEE 802.11a/n protocol in the physical layer. CSI describes the signal propagation and channel characteristics of the transmission link [34,35].

#### 2.1.1. Signal Transmission System Characterization

In the field of wireless communications, *H* describes the channel properties of a transmission link, which is a reflection of distance attenuation, shadow fading and multipath interference. The channel transmission model is expressed as follows:(1)$$\vec{Y}=H\cdot \vec{X}+\vec{N},$$
where $\vec{X}$ and $\vec{Y}$ respectively denote the transmitted and received signal vectors, $\vec{N}$ is the additive white Gaussian noise, and *H* is the channel frequency response.

#### 2.1.2. Multipath Channel Response in Time Domain

In a typical indoor environment, the transmitted signals travel through multiple paths to the receiving terminals. Each path introduces different amplitude attenuation, time delays, and phase shifts. In order to distinguish paths, the wireless channel can be modeled as a spatial linear filter to describe a channel impulsive response (CIR) [18]:(2)$$h\left(\tau \right)=\sum_{k=0}^{L_{p}-1}a_{k}\delta \left(\tau -\tau_{k}\right),$$
where $L_{p}$ denotes the number of multipaths, $a_{k}$ and $\tau_{k}$ represent the amplitude and time delay of the $k_{th}$ path respectively. By performing Inverse Fast Fourier Transform (IFFT) on CSI signal, we can obtain signal h(τ) in the time domain, which reflects signal strength at different time delays. The LOS path and multipath components can be then roughly distinguished. We set the threshold and filter the dropped pulses to obtain an approximate LOS path.

#### 2.1.3. Channel Response in Frequency Domain

According to the CSI definition, WLAN system based on OFDM uses multiple subcarriers for data transmission to reflect the frequency diversity. Channel conditions on each subcarrier include amplitude decay and phase shift. In narrow-band flat fading channel, the frequency domain OFDM system is modeled as:(3)$$H_{k}=\left\|H_{k}\right\|e^{i∠H_{k}},$$
where $H_{k}$ and $e^{i∠H_{k}}$ represent the amplitude and phase of the $k_{th}$ subcarrier respectively.

The CSI information obtained by the Intel Wi-Fi Link 5300 NIC contains three antenna information where each antenna contains 30 subcarriers from a received packet. The effective CSI of the packet is described as follows [18]:(4)$$CSI_{eff}=\frac{1}{k}\sum_{k=1}^{k}\frac{f_{k}}{f_{0}}\times \left|H_{k}\right|,k\in \left(-15,15\right),$$
where $f_{k}$ and $f_{0}$ indicate the frequency of the $k_{th}$ subcarrier and the central frequency of 30 groups of subcarriers respectively. $\left|H_{k}\right|$ is the amplitude of the $k_{th}$ subcarrier.

### 2.2. Challenges

Theoretically, the superiority of the specific wireless signal characteristics used for accurate positioning and target identification need to meet the following two aspects in an indoor scenarios.

CSI requires sufficient stability in a static environment. It should be provided with good anti-interference ability of corresponding frequency band signal.More importantly, we require an immediate response once the CSI is interfered. Consequently, the intrusion can be accurately identified in the environment covered by wireless devices. The degree of discrimination should be as subtle and sensitive as possible for different interference and environments.

However, simple Wi-Fi signal transceivers is always limited by hardware defects, and the achievement of high-precision positioning performance is a big challenge.

Next, the existing dynamic intrusion detection and LOS/NLOS discrimination are usually based on the setting of threshold. However, they may be confused and false alarmed when Intrusion and NLOS exist at the same time, which is also a challenge.

In addition, CSI is collected at considerable speed and hundreds of packets are transmitted even in one second. From the perspective of practical applications, it is also a great challenge to realize the immediate response of intrusion detection in the face of huge feature information.

We attempt to deal with the above three challenges through the pre-processing of amplitude and phase features, the introduction of B-SVC method for intrusion detection and an improved multi-class fingerprint positioning method, which will be explained in detail in Section 3.

## 3. Methodology

In this section, we introduce the design details of C-InP by actual measurements from two aspects—feature preprocessing for amplitude and phase separately and concrete methods used in C-InP.

### 3.1. Feature Preprocessing in C-InP

C-InP extracts CSI that conforms to IEEE 802.11 protocol as the original measurement observation.

#### 3.1.1. Improved Amplitude Outlier Filtering Method

To intuitively show the phenomenon that CSI amplitudes are location-specific in different LOS/NLOS conditions, we extracted and calculated CSI measurements from about 1000 packets in an antenna where the detailed experimental scenarios(spare room, single laboratory, integrated NLOS room, complex garage) and setup will be introduced in Section 4.1.

Figure 1, Figure 2, Figure 3 and Figure 4a illustrate that there are significant differences in CSI amplitudes between different locations and unique characteristic traits between LOS/NLOS conditions. In addition, the fact can also be observed that the original CSI limited by hardware defects is difficult to avoid measurement anomalies, which cannot be utilized directly for fingerprint location without feature preprocessing, especially in relatively complex scenarios. Figure 1, Figure 2, Figure 3 and Figure 4d show the corresponding boxplot diagrams for analyzing the amplitude outliers where the ranges between the upper and the lower quartiles are on order of magnitude of 1dB and 10dB separately in single scenes (Figure 1 and Figure 2) and complex scenes (Figure 3 and Figure 4). This challenge motivates us to explore probability distribution for further.

As can be observed in Figure 1, Figure 2, Figure 3 and Figure 4b, the different distribution characteristics of the amplitude probability distribution’s heat map in the above test environments have inspired us to utilize the probability distribution to implement outlier filtering. It can be concluded that probability distribution boundaries of simple LOS measurements are clear in the spare room and in the single laboratory, but blurry in the integrated NLOS room and the complex garage.

In order to eliminate amplitude outliers while retaining the specificity of features to the maximum, based on the analysis of the amplitude changes and distribution characteristics under different scenarios above, we propose a statistical matrix of median sequence measurement changes where outlier items are filtered according to the probability distribution of the variation. The measured CSI data forms the matrix *H* mentioned in Equation (Equation 1), with the size of $n_{a}\times n_{s}\times n_{p}$, where $n_{a}$, $n_{s}$ and $n_{p}$ are the number of antennas, subcarriers and data packets, respectively. The measured amplitude matrix is described as:(5)$$\begin{array}{c} H^{A}=\left[H_{1}^{A},H_{2}^{A},\cdots ,H_{n_{p}}^{A}\right],\\ H_{i}^{A}={\left[H_{1i}^{A},H_{2i}^{A},\cdots ,H_{(n_{a\times }n_{s})i}^{A}\right]}^{T}. \end{array}$$

Firstly, we select the median amplitude value $H_{jm}^{A}$ from $n_{p}$ packets:(6)$$H_{jm}^{A}=\left\{\begin{array}{c} \frac{H_{j}^{Am}\left(\frac{n_{p}}{2}\right)+H_{j}^{Am}\left(\frac{n_{p}}{2}+1\right)}{2},n_{p}\%2=0,\\ H_{j}^{Am}\left(\frac{n_{p}+1}{2}\right),n_{p}\%2=\frac{1}{2}. \end{array}\right.$$
where $H_{j}^{Am}=rank\left(\left[H_{j1}^{A},H_{j2}^{A},\cdots ,H_{jn_{p}}^{A}\right]\right)$ and *j* represents subcarrier index of the *j*th subcarrier. And we can then achieve the median sequence $H_{m}^{A}$ from the measured martix:(7)$$H_{m}^{A}={\left[H_{1m}^{A},H_{2m}^{A},\cdots ,H_{(n_{a\times }n_{s})m}^{A}\right]}^{T}.$$

Next, we calculate the deviation variance corresponding to each packet separately:(8)$$\begin{array}{c} \Sigma_{A}^{2}=\left[\sigma_{1}^{2},\sigma_{2}^{2},\cdots ,\sigma_{n_{p}}^{2}\right],\\ \sigma_{i}^{2}=\sum_{k=1}^{n_{a\times }n_{s}}{\left(H_{ki}^{A}-H_{km}^{A}\right)}^{2}. \end{array}$$

Figure 5 describes the cumulative distribution of variance between each packet and the median sequence in the above four environments. Finally, we delete entries that exceed the tolerance threshold of variation $\sigma_{thi}^{2}$, which can be determined by preliminary measurements in different measurement scenarios. Figure 1, Figure 2, Figure 3 and Figure 4c illustrate that the processed amplitudes remain the trend and specificity of origin features and the noises are removed effectively at the same time. The calibrated amplitude makes preparation for the establishment of the database for the next positioning step.

#### 3.1.2. Modified Phase LS calibration

In actual measurement process, the lack of hardware accuracy makes the true phase difficult to obtain. Previous studies [36] explicitly point that phase measurement error comes from two major reasons—Carrier Frequency Offset (CFO) and Sampling Frequency Offset (SFO). The former is caused by the fact that the center frequency of the transmitting and receiving pairs cannot be completely synchronized, resulting in phase rotation error [37]. The latter may cause the received signal after ADC a time shift with respect to the transmitted signal [38]. The linear transformation is usually proposed to minimize the CFO and SFO that are not fully compensated.

CSI provides phase measurement information for each subcarrier, the measured phase $\overset{\wedge }{\phi_{k}}$ of the $k_{th}$ subcarrier can be expressed as:(9)$$\overset{\wedge }{\phi_{k}}=\phi_{k}-\frac{2\pi f_{k}}{N}\delta +\beta +Z,$$
where $\phi_{k}$ is the true phase, δ denotes time offset of the receiver, which is the main reason for the phase error in the middle frequency band. $f_{k}$ indicates the $k_{th}$ subcarrier index and N is the FFT size. According to IEEE 802.11n, $f_{k}$ is ranging from −28 to 28 and N equals to 64. β represents the phase offset and *Z* is the random noise.

Since each packet has a different time delay, which appears as a slope of the phase-frequency characteristic curve, the linear transformation is performed for each packet separately. Inspired by the analysis of amplitude characteristics, the discussion of phase calibration is mainly divided into simple LOS and complex NLOS conditions. Figure 6a and Figure 7a illustrate the original phase and the phase eigenvectors after linear transformation of 30 subcarriers on the same antenna in LOS/NLOS conditions, respectively. Unfortunately, the phase frequency characteristics cannot completely guarantee the ideal linear characteristics under the condition of the same time delay. The phase stagnation of adjacent subcarrier may generated in the case of NLOS, resulting in a serrated mutation in the phase frequency curve. Therefore, the conventional calibration method as recommended in Reference [34] may produce significantly deviation from the true phase. In order to mitigate the impact of mutation, the Least Squares (LS) method is taken into consideration, as Equation (Equation 10) shows.
(10)$$\left(\hat{w},\hat{b}\right)=arg\min\sum_{k=1}^{n}{\left(\overset{\wedge }{\phi_{k}}-wf_{k}-b\right)}^{2}.$$

$\left(\hat{w},\hat{b}\right)$ denotes the slope and intercept of the fitting line. On the basis of sacrificing the deviation between a few low-frequency subcarriers and their real phase, the effect of the serration mutation of the remaining subcarriers is compensated in this way. We obtain a linear combination of calibrated phase $\tilde{\phi_{i}}$ denotes the true phase $\phi_{k}$:(11)$$\tilde{\phi_{i}}=\overset{\wedge }{\phi_{i}}-\hat{w}k_{i}-\hat{b}.$$

As shown in Figure 6b and Figure 7b, we intuitively obtain a relatively stable distribution after the modified phase LS calibration in the perspective of continuously measured packets under a single subcarrier. Compared with the traditional linear transformation method, Figure 6c,d and Figure 7c,d shows that the proposed LS calibration method has significantly optimized the measured phase, especially in NLOS situations with about average 75% performance improvement in redundant variance range of each subcarrier. As observed in Figure 6b, the conventional and modified calibration methods are not much different in the case of LOS, and the distribution range of each subcarrier is strictly controlled at 0.05 radians.

Moreover, Figure 8 shows the calibration phase of four different positions respectively, showing a clear distinction as expected. Therefore, the processed phase features can be classified as feature values, which is similar to the rationale for amplitude-based features. Figure 9b depicts the variation in phase difference in the LOS/NLOS scenarios. The performance of the processed CSI amplitude and phase in the positioning improvement will be evaluated in Section 4.3.3.

### 3.2. C-InP—Indoor NLOS Positioning Based on Enhanced CSI Feature with Intrusion Adaptability

#### 3.2.1. System Architecture of C-InP

Firstly, We provide an overview of C-InP and analyze the detection and positioning operating procedures briefly. The system architecture is provided in Figure 10. In terms of amplitude and phase, Figure 11 illustrates the time stability of 3×30 subcarriers in 60,000 packets collected without subjective interference in LOS path during the whole day. It indicates that the fluctuation caused by time is much smaller than that between subcarriers, thus we regard the amplitude and phase characteristics as available.

Let us start with a hypothetical scenario—A target starts walking in the detection area and the collected CSI samples are constantly analyzed by a detection indicator to determine if intrusion exists or not. This step requires feature pre-processing for more accuracy. Once the indicator is YES, the CSI samples are further fed into the position estimation module. The two components of C-InP are briefly introduced as follows:

***(1) Intrusion Detection:*** The key to intrusion detection is to achieve real-time response. The measured CSI characteristics between the transmitter and receiver changes once intrusion appears in search area. We extract the eigenvalue vector of the correlation feature matrix of amplitude and phase to describe the current signal. In practical applications, CSI is collected at considerable speed [20], so we set a fixed time window and judge the dynamic intrusion based on the signal changes between adjacent time windows.

***(2) Position Estimation:*** The key to position estimation is to construct efficient mapping relationship between raw features and the corresponding positions. This module will be triggered when a target is detected. CSI data exhibits significant pattern differences for targets at different locations. Such property of CSI makes it possible to achieve positioning through classification. According to the mapping relationship, we use the multi-classification method to maximize the difference between the reference points for accurate indoor localization. More details will be explained in the following subsections.

#### 3.2.2. Binary-SVC for Intrusion Detection

The setting of detection module is to alleviate the computation burden of the practical positioning module. As is well-known, CSI is collected at considerable speed and hundreds of packets are transmitted even in one second. From the perspective of practical applications, we set a fixed time window and judge the dynamic intrusion based on the signal changes between adjacent time windows in detection module.

Inspired by the amplitude Outlier filtering process in Section 3.1, we explore the effect of NLOS propagation on CSI signals. Figure 9 depicts the variation in phase difference in the LOS/NLOS scenarios. NLOS transmission has brought about large fluctuations, which tend to be confused as intrusions. Consequently, feature variation of envelope variance is too optimistic to satisfy the accurate detection. Aiming at this problem, C-InP adopt support vector machine (SVM) for binary classification to detect whether intrusion or not, and the Binary Support Vector Classification (B-SVC) model is established.


*(1) Classification model*


The B-SVC model of a linear kernel function is a linear combination of feature-based features. Training sample consists of a pair $\left(H_{csi}^{{}^{\left(i\right)}},y^{\left(i\right)}\right)$, where $H_{csi}^{{}^{\left(i\right)}}$ represents the sample feature and $y^{\left(i\right)}$ refers to the classification label. For binary classification problem, $y^{\left(i\right)}\in \left\{1,-1\right\}$, which indicates intrusion positive and negative separately.

An efficient way to solve binary classification problems is to search for a dividing hyperplane in the feature space, which can separate the samples belonging to different classification tags. The hyperplane need to meet the following conditions:(12)$$y^{\left(i\right)}\left(w^{T}H_{csi}^{{}^{\left(i\right)}}+b\right)>0.$$

In order to obtain higher tolerance for random disturbances, a hyperplane with the largest geometric interval is the key optimization target to improve the discrimination and classification performance. Suppose that there is a point $p\in R^{d}$ and a hyperplane $w^{T}x+b=0$, and the distance between them can be expressed as:(13)$$d_{P}=\frac{1}{∥w∥}\left|w^{T}p+b\right|,$$
where w and *p* denote the normal vector of the hyperplane and the coordinate vector of the target point respectively, and *b* is a constant term.


*(2) Optimization model simplification*


For constrained optimization problems, it is generally solved by constructing a Lagrange function as follows:(14)$$L\left(w,b,\alpha \right)=\frac{1}{2}w^{T}w+\sum_{i=1}^{m}\alpha_{i}\left(1-y^{\left(i\right)}\left(w^{T}H_{csi}^{{}^{\left(i\right)}}+b\right).\right)$$

The above formula must satisfy the KKT condition [39] at the optimal value. The use of dual problems can better solve the solution and have higher computational efficiency. Therefore, under the Slater condition [40], the optimal solution of Equation (Equation 14) is equivalent to the dual problem:(15)$$\begin{array}{c} \max_{\alpha }\min_{w,b}\frac{1}{2}w^{T}w+\sum_{i=1}^{m}\alpha_{i}\left(1-y^{\left(i\right)}\left(w^{T}H_{csi}^{{}^{\left(i\right)}}+b\right)\right)\\ s.t.\;\alpha_{i}\geq 0,\;i=1,2,\cdots ,m \end{array}$$.

When the partial derivative of the Lagrange function (Equation (Equation 14)) is equal to 0, the optimal solution $\left(w,b\right)$ can be obtained.
(16)$$\frac{\partial L}{\partial w}=0\Rightarrow w=\sum_{i=1}^{m}\alpha_{i}y^{\left(i\right)}H_{csi}^{{}^{\left(i\right)}},$$
(17)$$\frac{\partial L}{\partial b}=0\Rightarrow \sum_{i=1}^{m}\alpha_{i}y^{\left(i\right)}=0.$$

Finally, Equations (Equation 15)–(Equation 17) are combined to eliminate and $\left(w,b\right)$ then a set of parameters is $\alpha_{i}$ obtained by using the quadratic programming. Accordingly, the optimized hyperplane can be solved out.
(18)$$\begin{array}{c} \;\min_{\alpha }\frac{1}{2}\sum_{i=1}^{m}\sum_{j=1}^{m}\alpha_{i}\alpha_{j}y^{\left(i\right)}y^{\left(j\right)}H{{}_{csi}^{{}^{\left(i\right)}}}^{{}^{T}}H_{csi}^{{}^{\left(j\right)}}-\sum_{i=1}^{m}\alpha_{i}.\\ \;s.t.\;\sum_{i=1}^{m}\alpha_{i}y^{\left(i\right)}=0,\;i=1,2,\cdots ,m \end{array}.$$

Since the high-dimensional features of different classes are not necessarily linearly separable, we choose Radial Basis Function (RBF) as the kernel function to map the original features to the high-dimensional space ($\phi :R^{d}\to R^{\tilde{d}}$).
(19)$$\phi \left(H_{csi}^{{}^{\left(i\right)}},H_{csi}^{{}^{\left(j\right)}}\right)=e^{\left(-\frac{{∥H_{csi}^{{}^{\left(i\right)}}-H_{csi}^{{}^{\left(j\right)}}∥}^{2}}{2\sigma^{2}}\right)}.$$

The parameter $\sigma^{2}$ trained for classification is determined by cross-validation. In addition, adjusting the kernel function can transform features from low to high dimensions to improve feature discrimination and achieve better classification performance.

#### 3.2.3. Improved Multiple-SVC Method for Fingerprint Localization


*(1) CSI channel matrix singular value decomposition(SVD)*


Eigenvalue decomposition (EVD) is a common method for extracting the main features of a matrix, but its application is limited to square matrix. As for general matrices, we take singular value decomposition (SVD) into consideration. According to the principle of SVD, the channel state matrix explained in Equation (Equation 1) can be decomposed into the following Equation (Equation 20):(20)$$CSI_{m\times n}=U_{m\times m}\Lambda_{m\times n}V_{n\times n}^{T},,$$
where Λ is the diagonal eigenvalues matrix of CSI, $\Lambda =diag\left(\lambda_{1},\lambda_{2},\cdots ,\lambda_{n}\right)$ arranged in descending order. The column vector of *V* represents the standard orthogonal basis of the original domain whereas *U* refers to the new standard orthogonal basis after transformation. Λ denotes the scaling relationship between the column vectors in *V* and in *U*.

The premise of utilizing SVD for data information extraction or data dimension compression is to preserve the most significant information in the matrix. We directly extract the first *k* terms that account for 99% of the energy information of the matrix. Since the sequence of eigenvalues $\left(\lambda_{1},\lambda_{2},\cdots ,\lambda_{n}\right)$ is arranged in descending order, the solution formula for k is:(21)$$k=\arg\min_{k}\frac{\sum_{i=0}^{k}\lambda_{i}^{2}}{\sum_{i=0}^{n}\lambda_{i}^{2}}\geq 99\%,$$

We obtain *k* largest eigenvalues $\Lambda^{\prime }=diag\left(\lambda_{1},\lambda_{2},\cdots ,\lambda_{k}\right)$. Meanwhile, the corresponding eigenvectors in *U* and *V* is constructed as $U^{\prime }$ and $V^{\prime }$. Then we have a processed CSI matrix after redundancy information processing:(22)$$CSI_{m\times n}^{\prime }=U_{m\times k}^{\prime }\Lambda_{k\times k}^{\prime }{V_{n\times k}^{\prime }}^{T}.$$

According to the above inference, a new m×k dimensional observation $H_{m\times k}$ is formed and signal to Noise Ratio (SNR) is then greatly improved while maintaining the uniqueness of the eigenvalue. The low-dimensional matrix is implemented to approximate the original m×n dimensional matrix.
(23)$$H_{m\times k}=CSI_{m\times n}^{\prime }{{V_{n\times k}^{\prime }}^{T}}^{-1}=U_{m\times m}\Lambda_{m\times k}^{\prime }.$$


*(2) M-SVC model*


In order to realize the consistency of the detection and positioning methods of the C-InP, on the basis of the above B-SVC model, an improved multiple classification method is extended for indoor localization. As for *k* classification samples $\left\{\left(w_{1},b_{1}\right),\left(w_{2},b_{2}\right),\cdots ,\left(w_{k},b_{k}\right)\right\}$, which are extracted from the above low-dimensional matrix $H_{m\times k}$, it is required to design a binary classification process for $C_{k}^{2}$ times. In the final Multiple-SVC model, the results of the correctly labeled class are higher than the results of other classes at the interval of γ=1 [41], The form is as follows:(24)$$\begin{array}{c} \min_{W,b}\frac{1}{m}\sum_{i=1}^{m}\sum_{k=1}^{K}\max(0,f\left(i,y_{i}\right)-f\left(i,k\right)+1)+\frac{\lambda }{2}\sum_{k=1}^{k}w_{k}^{T}w_{k},\\ \;f\left(i,t\right)=(w_{t}^{T}\phi \left(H_{csi}^{{}^{\left(i\right)}}\right)+b_{t}). \end{array}$$

The performance of the improved M-SVC method for localization in the positioning improvement will be evaluated in Section 4.3.

## 4. Experiments and Performance Evaluation

In this section, we interpret the experimental setup, evaluation metrics and detailed performance evaluation. The experimental setup contains the implementation of CSI data collection and experimental scenarios two parts. The performance of the proposed C-InP is evaluated and discussed in two aspects—detection and positioning. The comparison of the proposed preprocessing method is introduced firstly. The effect of different parameter K in SVD is then evaluated, and finally the positioning system performance of C-InP is presented and compared with one MDS-KNN approach and one Naive Bayes (NB) approach.

### 4.1. Experimental Setup

#### 4.1.1. CSI Data Collection

As shown in Figure 12 and Figure 13, we assembled and configured our own signal transmitters and receivers, which are equipped with built-in commercial Intel 5300 NIC and Ubuntu14.04 operating system. According to the IEEE 802.11n protocol, the acquisition of CSI features is achieved by Linux 802.11n CSI Tool [20] and operating in monitor mode at 2.4 GHz at the intervals of 5 ms. The number of antennas at the transmitters and the receivers are 1 and 3 respectively. We leverage libsvm development kit [42] to achieve classification. The experiment is carried out in three typical indoor scenarios, and the detailed area division layout is as shown in the Figure 14.

#### 4.1.2. Experimental Scenarios

We implement experiments to show the performance of our system in three different scenarios in the campus of Beijing University of Posts and Telecommunications as follows:*Spare room:* This scene is characterized by the fact that the transmission links are mostly LOS paths, which is beneficial for better comparison of system performance in other scenarios.*Integrated NLOS room:* Specially, we take NLOS propagation into consideration. We deploy the experiment in a integrated room environment of 8 m × 16.5 m, which contains a single laboratory (scene 2 mentioned in Section 3) and a meeting room and is separated by a glass wall, as shown in Figure 14a. Due to the complex indoor environment, we place four receiving terminals with a height of about 2.3 m to ensure the transmission of CSI signals. The receivers are respectively fixed at the reference category shown in Figure 15. The transmitter then moves progressively along a predetermined route maintaining a height of 1.4 m.*Complex garage:* In addition, we also selected an underground parking garage to verify system performance, as shown in Figure 14b. In order to adapt to the relatively large area of the environment, the interval between every two reference points is 1.5 m which is larger than other two scenes. As a representative experimental scenario, the impact of basement environment on experimental performance can be further explored.

The major parameters of the above three scenarios are introduced in Table 1. We measure two categories for comparison on each transmission link—the situation when someone invades the transmission link; the situation where the environment is relatively static, where each category is measured for two times. For the sake of fairness, we collect a similar amount of CSI sampling data cells for every reference category. There are several students sit or walk outside the union area of the Fresnel zone with TX-RX as the focus to show the robustness of C-InP. In order to control the experimental environment variables more scientifically, we take advantage of the remote control terminal to conduct our experiments.

### 4.2. Evaluation Metrics

This paper uses the following metrics as the evaluation criteria to evaluate the experimental detection results.

True Positive (TP)—the ratio of instances where an existed intrusion is correctly detected.True Negative (TN)—the fraction of cases where no intrusion presence is correctly identified.Accuracy—the probability of making the right judgment about the invasion and vacancy.

(25)$$Accuracy=\frac{TP+TN}{TP+FP+TN+FN},$$
where FP and FN stand for the “False Positive” (equals to 1-TP) and “False Negative” (equals to 1-TN) respectively. In order to compare with different positioning approaches, we use the mean distance error (MDE) as a performance indicator and separately plot cumulative error probability of each method. The estimated positions is represented by $X_{obj,i}$, and the actual position is described as $X_{act,i}$, so the representation of N positioning points for:(26)$$MDE=\frac{1}{N}\sum_{i=1}^{N}∥X_{obs,i}-X_{act,i}∥$$
(27)$$X_{obs,i}=\left(\overset{\wedge }{x_{i}},\overset{\wedge }{y_{i}}\right),X_{act,i}=\left(x_{i},y_{i}\right)$$

### 4.3. Performance Evaluation

#### 4.3.1. Performance of Detecting Intrusion

According to the measurement of two categories for each transmission link of three scenarios described in Section 4.1.2, we first evaluate the intrusion detection module of the system and integrate statistical detection accuracy, where we set 1s as the fixed time window length after comprehensive consideration.

Figure 16a shows the single-person intrusion performance of the detection module in three scenarios. The detection accuracy of our system can reach 98.2%, 89.4% and 94.7% respectively in the spare room, the integrated room and the complex garage. It indicates that NLOS interference seriously affects the classification results. In addition, we compare B-SVC used in C-InP with the Naive Bayesian algorithm [19], as shown in Figure 16b. We can observe that the accuracy of the two approaches in the spare room is not much different. However, the detection accuracy of the former is about 14.75% and 9.1% higher than the latter in the integrated room and the complex garage, respectively. The results indicate that B-SVC is more effective in NLOS scenarios. Nevertheless, the Naive Bayesian algorithm has a certain advantage in time performance.

#### 4.3.2. Performance of Indoor Localization

Next, we evaluate the positioning performance of the C-InP system using improved M-SVC in NLOS scenarios. As is elaborated in Figure 17, the positioning accuracy in the integrated room and complex garage environments is 92.6% and 84.6%, respectively, and the corresponding MDE is 0.49 m and 0.81 m. Within 2 m of the distance error, the accuracy is respectively 93.1% and 87.5% in the integrated room and complex garage. In contrast, the positioning accuracy and MDE of the garage is lower than in a integrated room laboratory environment. We suspect that this is because the garage environment cannot guarantee the transmission link not disturbed by the vehicle during the transmission process. In addition, the impact of human objective factors on positioning results is inevitable in actual implements. Moreover, it is positioned to another class with at least an interval of 1.2 m/1.5 m once it is misjudged. Therefore, the maximum distance error is relatively large.

#### 4.3.3. Comparison of Different Preprocessing Methods for Both Amplitude and Phase

In order to demonstrate the effectiveness of the preprocessing methods, the typical DBSCAN algorithm is used for fair comparison to evaluate the positioning accuracy in two typical indoor NLOS environments. Figure 18 illustrates the processing results of the C-InP, the DBSCAN algorithm as well as the raw data.

In general, the preprocessing method in C-InP is superior to the above two methods. In the integrated room environment, Figure 18a shows that C-InP can ensure 93.1% positioning accuracy with the error distance of less than 2 m, while DBSCAN accounts for 91.0% to obtain the positioning error under 2 m. The accuracy of the raw data classified by the improved M-SVC is approximately 86.3% within the error of 2 m. As shown in Figure 18b, C-InP provides the positioning accuracy of 84.6% in the complex garage environment, higher than DBSCAN and the raw data about 1.6% and 10%, respectively. All positioning accuracy and MDE results are illustrated in Table 2.

#### 4.3.4. Effect of Different Parameter K in SVD for the Improved Multiple-SVC Method

To compare the effects of the parameter *k* in the SVD, Figure 19 depicts the CDF plots for different k values in the above two scenes, where the maximum error distance without SVD is larger than the processed. We observe that the error is smaller in the case of k=30 than other cases in Figure 19a, and the error gradually accumulates below this value. Particularly, the features can still be accurately distinguished even if the retained dimension is only 6, accounting for 6.67% of the original. Compared with the complex garage, as shown in Figure 19b, the positioning effect is best when *k* = 35, which is a little larger than that in the integrated room. We suspect this is because the measurements in garage are greatly affected by inevitable body factors, so there are a little more number of eigenvalues that lead to its dominant feature. In summary, *k* is critical in order to both reduce noise and maintain feature uniqueness. In general, we extract the *k* that the corresponding energy accounts for 99% of the original feature when both taking the complexity of the operation and positioning accuracy into consideration.

#### 4.3.5. Performance Comparison of Different Positioning Systems

The system performance evaluation is mainly divided into two aspects—positioning accuracy and computational complexity. We compare our work with previous work MDS-KNN [22] and Naive Bayes(NB) [19]. MDS-KNN used KNN algorithm to locate and utilized MDS to reduce computational complexity, while the other is a Naive Bayes classifier-based technique. Figure 20 shows that the positioning accuracy of our proposed system is 15.2% and 29.4% higher than MDS-KNN and NB, respectively. Table 3 indicates the computational complexity of different systems and we use Intel(R) i3-4150CPU as the processor with 8GB memory. From the location-only system perspective, improved M-SVC’s average run-time is slightly longer than that of the others. The intrusion module design of C-InP improved the system utility performance by about 19% higher than the original. Nevertheless, MDS-KNN is more suitable for NB-IoT system, while NB is a probabilistic classification algorithm which is fitter for the case where the test point is far from the reference point.

### 4.4. Discussion of the Proposed Methods in C-InP

In this section, comprehensive experimental verification is carried out to evaluate the performance of the proposed methods in typical indoor scenarios and comparison results show that C-InP outperforms the existing systems in both LOS/NLOS environments from the perspective of detection accuracy and the corresponding MDE. We evaluated the performance of the proposed C-InP in two aspects—detection and positioning accuracy improvements contributed by the novel feature preprocessing methods and the improved classification methods of C-InP.

In the aspect of the novel feature preprocessing improvement, we evaluated the positioning accuracy performance of the improved amplitude outlier filtering method and the modified phase LS calibration used in C-InP, as mentioned in Section 3.1, are superior to that of the conventional DBSCAN algorithm, which has been proved in Section 4.3.3.

With regard to the improved methods of C-InP, we firstly evaluated the improved detection performance of the proposed C-InP in two typical scenarios. As has been proved in Section 4.3.1, the binary-SVC model utilized in C-InP outperformed the Naive Bayes algorithm for confused intrusion detection under NLOS conditions. Next, we evaluate the overall positioning performance of the C-InP using improved M-SVC in NLOS scenarios in Section 4.3.2. In addition, we also evaluated effect of the parameter *k* in SVD used in Section 3.2.3 with the conclusion that *k* accounts for 99% of the corresponding energy of the original feature, which has been proved in Section 4.3.4. Finally, the results shown in Section 4.3.5 have proved from the experimental aspect that the optimized SVD-based fingerprint method and the integrated establishment of intrusion detection module could actually bring positioning accuracy improvement and computational complexity reduction for integrated system C-InP with the comparison of one MDS-KNN approach and one NB approach. It is concluded that the MDE reached 0.49 m in the integrated room and 0.81 m in the complex garage, respectively.

Nevertheless, C-InP still retains some shortcomings. For example, we did not perform positioning tests at the position deviating from the reference point. In the future, we intend to further alleviate the computational complexity and try multi-target detection and positioning tracking.

## 5. Conclusions

In this paper, we propose a practical indoor positioning system based on enhanced CSI feature with pre-intrusion detection adapted to NLOS scenarios. In order to realize better feature resolution under NLOS conditions, we preprocess characteristics comprehensively with improved calibration methods for amplitude filtration and modified phase transformation. In addition, the integrated establishment of intrusion detection module and optimized SVD-based fingerprint localization module breaks through the limitations of indoor NLOS scenarios for high-resolution positioning. The binary-SVC model in C-InP is utilized for confused intrusion detection under NLOS conditions and an improved multiple-SVC model is established to realize low-complexity for indoor localization. Extensive experiments are conducted for evaluations and comparisons with different methods verifying the effectiveness of C-InP. The comparison of the proposed preprocessing method is discussed with the conventional DBSCAN algorithm. It is concluded that the improved amplitude outlier filtering method and the modified phase LS calibration used in C-InP is superior. The effect of different parameter *k* in SVD is also evaluated with the conclusion that *k* accounts for 99% of the corresponding energy of the original feature. The positioning performance of C-InP is finally presented and compared with one MDS-KNN approach and one NB approach. On the basis of improving the practicability of the location-only system by about 19%, the detection accuracy is guaranteed by 92.6% and 84.6% respectively in two scenes, as well as the corresponding MDE is 0.49 m and 0.81 m.

## Figures and Tables

**Figure 1 sensors-20-01211-f001:**
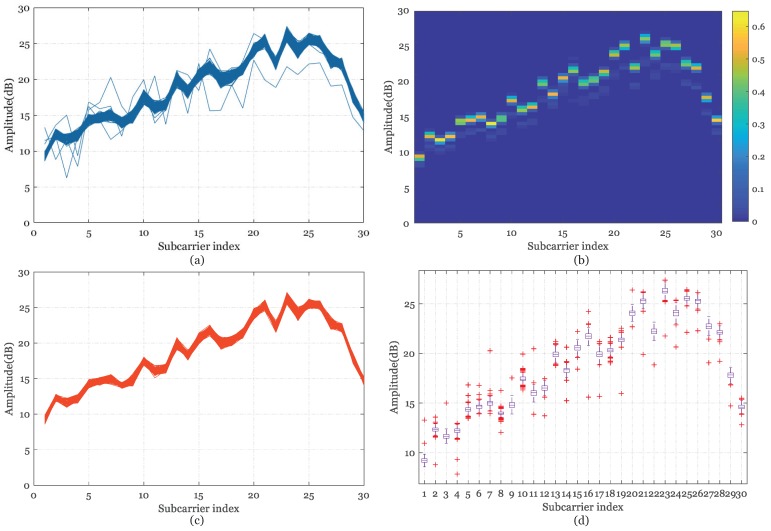
Channel state information (CSI) amplitude outlier filtering processing at position 1 in scenario 1 (Spare room). (**a**) The Raw CSI amplitude. (**b**) The distribution heat map of the amplitude probability. (**c**) The Processed CSI amplitude. (**d**) The boxplot diagram of the measured CSI amplitude.

**Figure 2 sensors-20-01211-f002:**
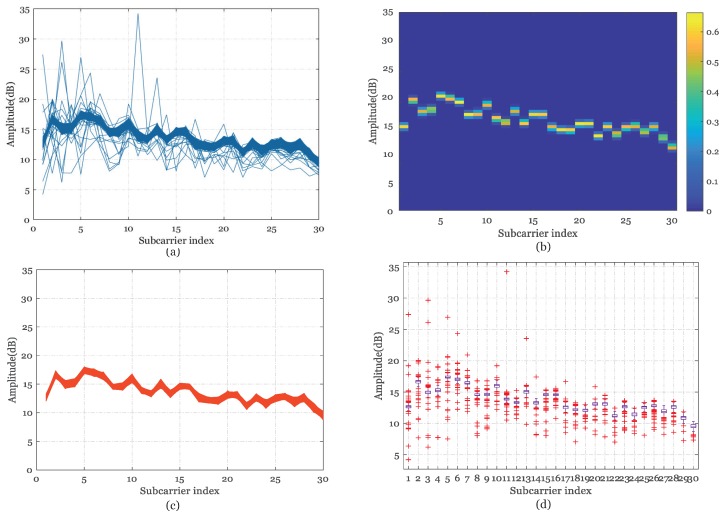
CSI amplitude outlier filtering processing at position 2 in scenario 2 (Single laboratory). (**a**) The Raw CSI amplitude. (**b**) The distribution heat map of the amplitude probability. (**c**) The Processed CSI amplitude. (**d**) The boxplot diagram of raw CSI amplitude.

**Figure 3 sensors-20-01211-f003:**
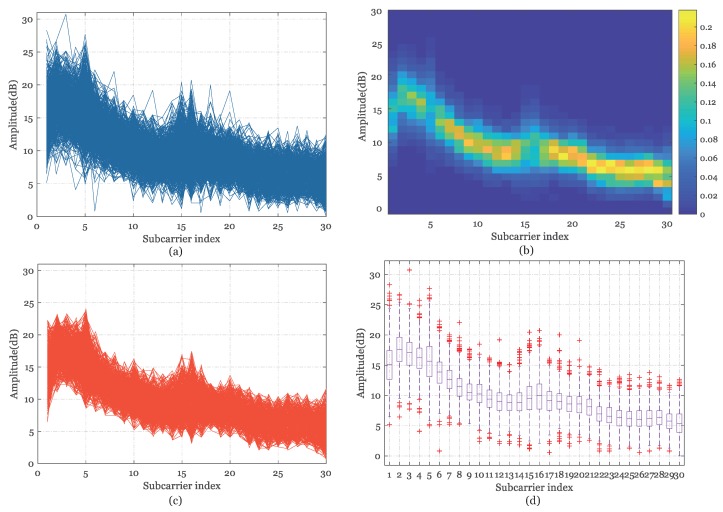
CSI amplitude outlier filtering processing at position 3 in scenario 3 (Integrated non line of sight (NLOS) room). (**a**) The Raw CSI amplitude. (**b**) The distribution heat map of the amplitude probability. (**c**) The Processed CSI amplitude. (**d**) The boxplot diagram of raw CSI amplitude.

**Figure 4 sensors-20-01211-f004:**
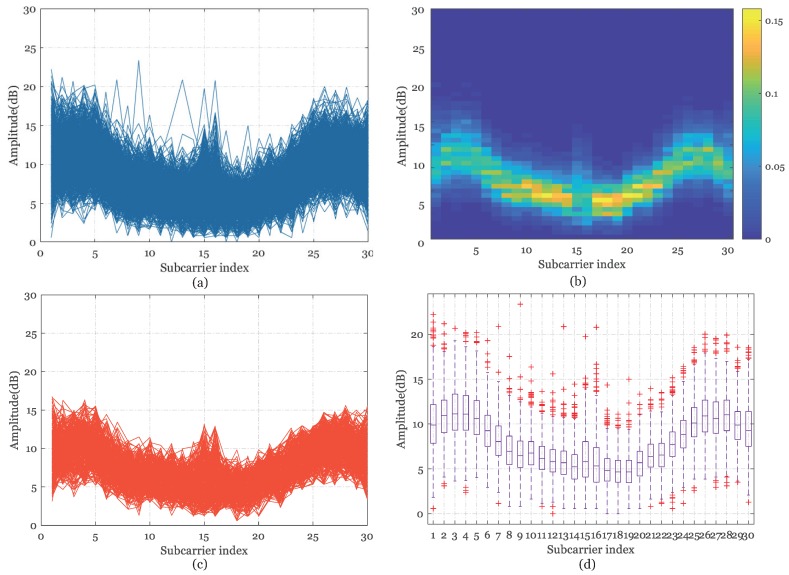
CSI amplitude outlier filtering processing in scenario 4 (Complex garage). (**a**) Raw CSI amplitude at position 4. (**b**) The distribution heat map of the amplitude probability at position 4. (**c**) Processed CSI amplitude at position 4. (**d**) The boxplot diagram of raw CSI amplitude at position 4.

**Figure 5 sensors-20-01211-f005:**
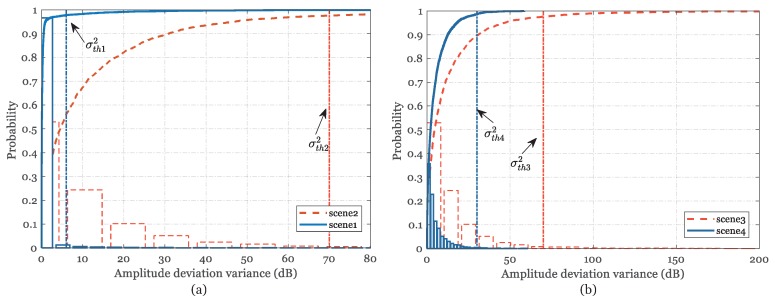
Amplitude deviation variance in different scenes. (**a**) Raw CSI amplitude distribution in single line of sight (LOS) conditions (scene1 & scene2). (**b**) Raw CSI amplitude distribution in complex conditions (scene3 & scene4).

**Figure 6 sensors-20-01211-f006:**
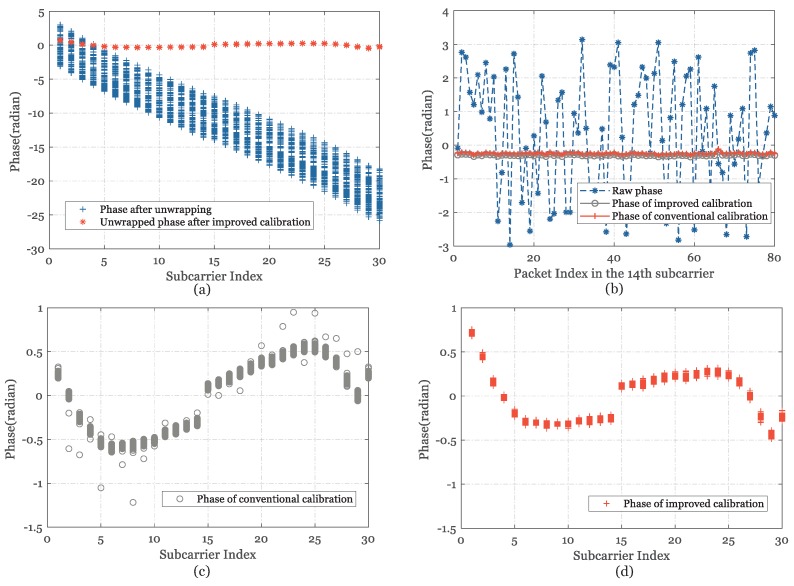
CSI phase calibration at LOS position. (**a**) CSI phase in one antenna. (**b**) CSI phase after linear transformation in the first subcarrier. (**c**) CSI phase after conventional calibration. (**d**) CSI phase after modified calibration.

**Figure 7 sensors-20-01211-f007:**
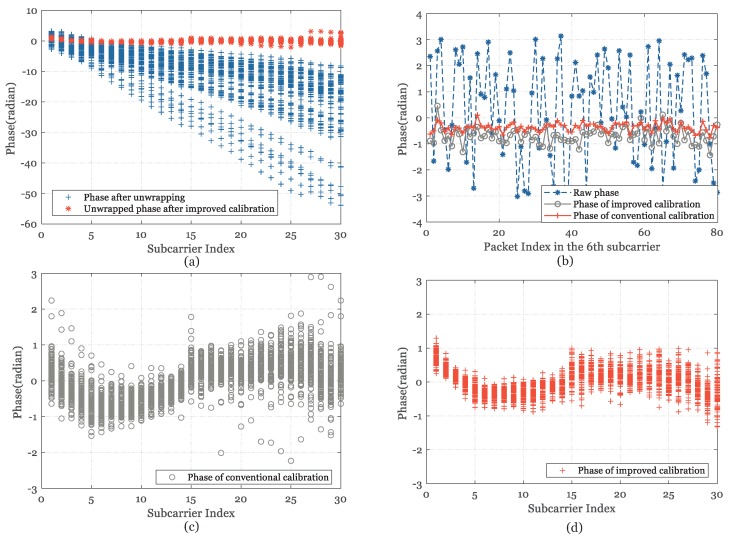
CSI phase calibration at NLOS position. (**a**) CSI phase in one antenna. (**b**) CSI phase after linear transformation in the first subcarrier. (**c**) CSI phase after conventional calibration. (**d**) CSI phase after modified calibration.

**Figure 8 sensors-20-01211-f008:**
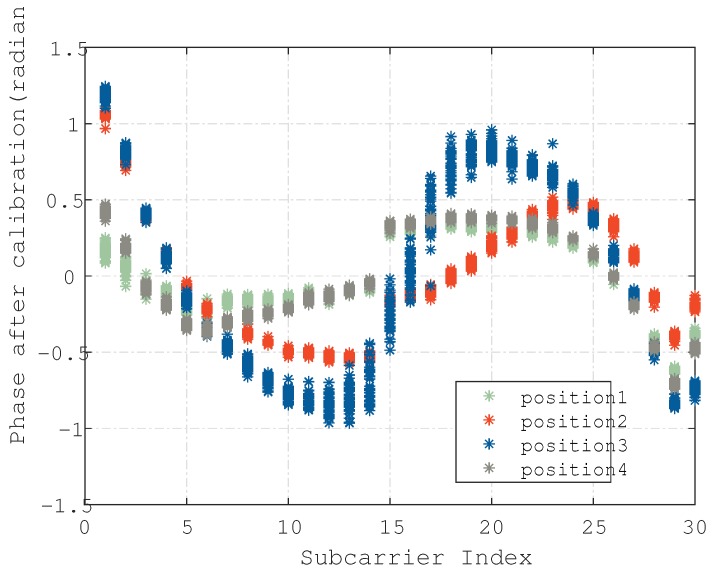
Phases in four different positions on the first antenna.

**Figure 9 sensors-20-01211-f009:**
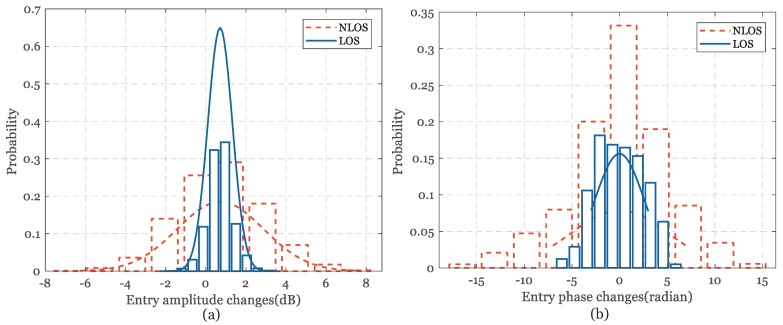
Temporal variances of CSI in LOS and NLOS. (**a**) CSI amplitude variances between adjacent entries. (**b**) CSI phase variances between adjacent entries.

**Figure 10 sensors-20-01211-f010:**
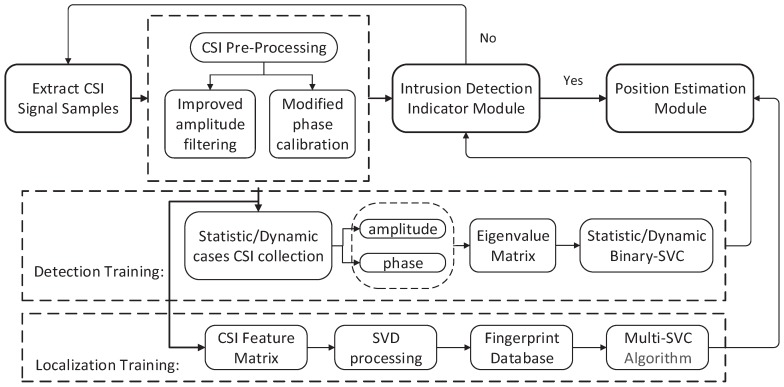
System architecture of C-InP.

**Figure 11 sensors-20-01211-f011:**
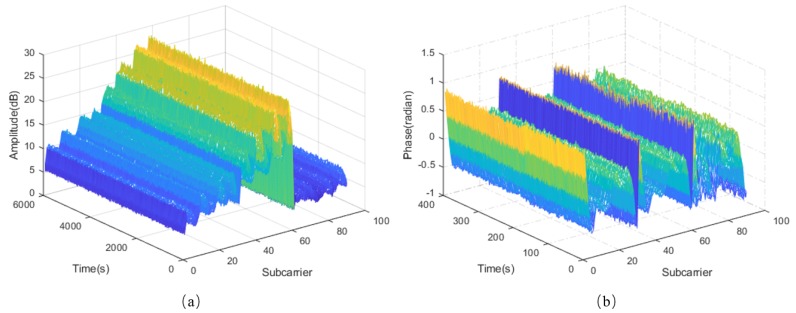
Time stability analysis (**a**) CSI Amplitude. (**b**) CSI Phase.

**Figure 12 sensors-20-01211-f012:**
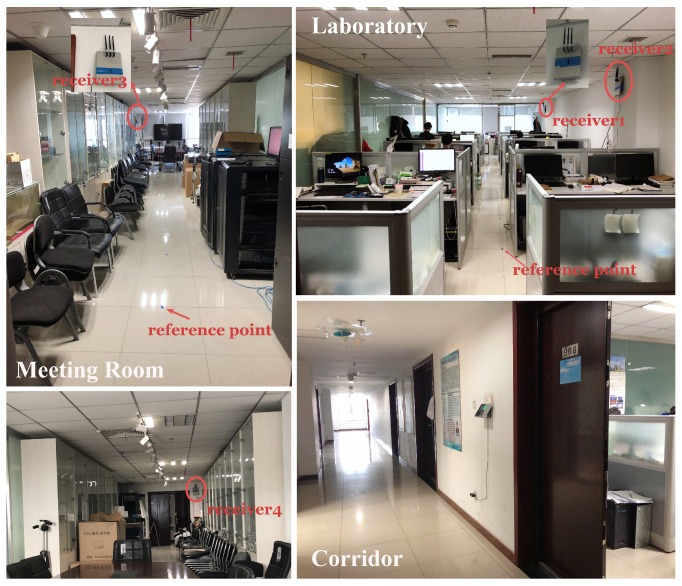
Experimental environments of the integrated NLOS room combined with laboratory, meeting room and corridor.

**Figure 13 sensors-20-01211-f013:**
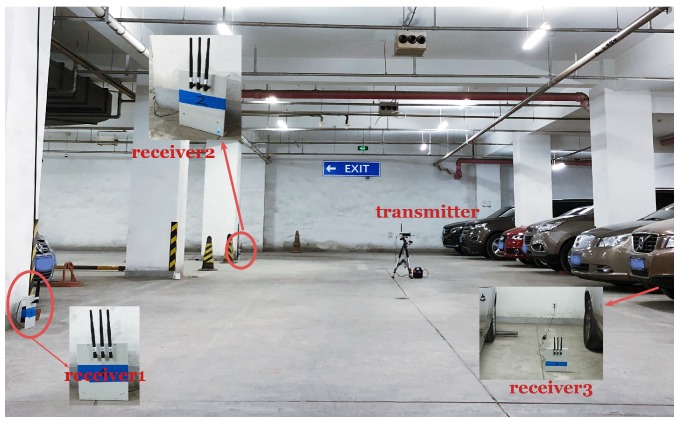
Experimental environments of the complex garage.

**Figure 14 sensors-20-01211-f014:**
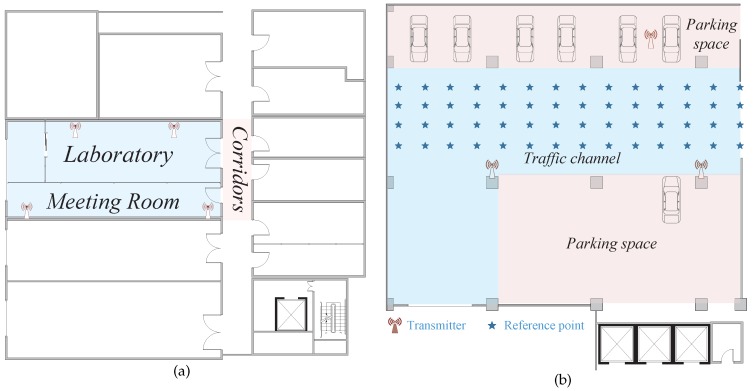
Experimental scenarios. (**a**) Integrated NLOS room. (**b**) Complex garage.

**Figure 15 sensors-20-01211-f015:**
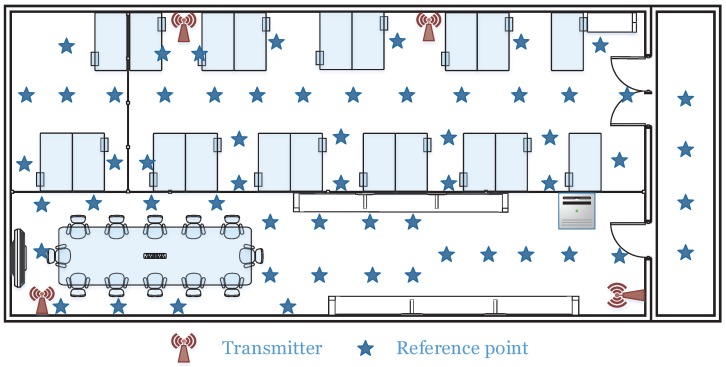
Detailed integrated room environment.

**Figure 16 sensors-20-01211-f016:**
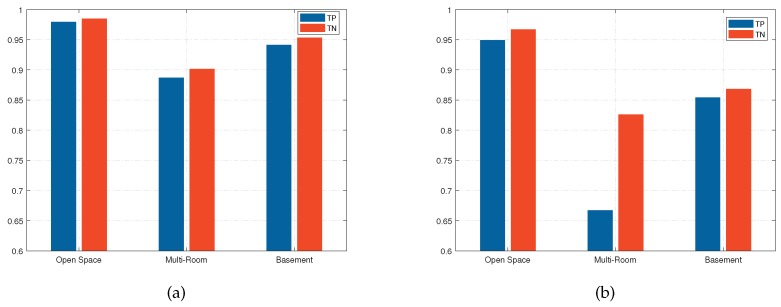
Detection results. (**a**) B-SVC. (**b**) Naive Bayes Classification.

**Figure 17 sensors-20-01211-f017:**
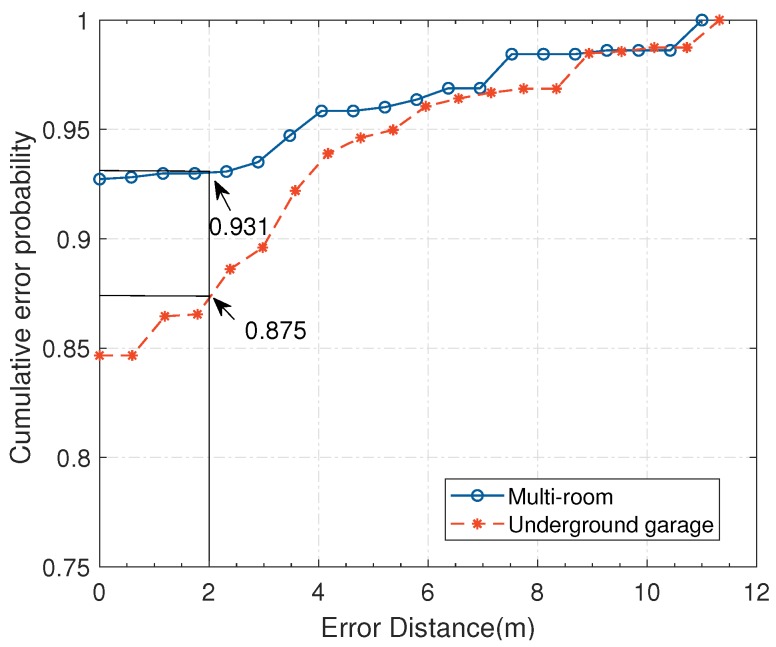
The Cumulative Distribution Function (CDF) of the error distance of C-InP in two scenes.

**Figure 18 sensors-20-01211-f018:**
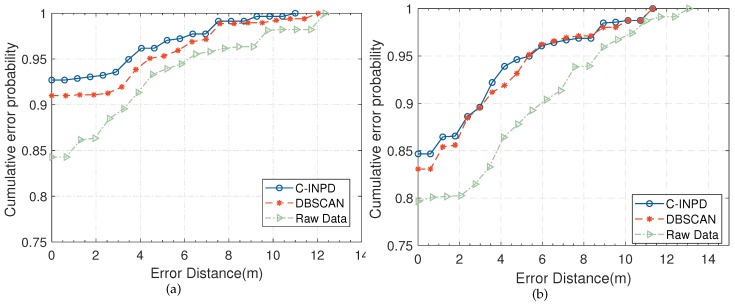
The comparison of positioning performance among C-InP, DBSCAN and raw data-based approach. (**a**) The CDF comparison in the integrated room. (**b**) The CDF comparison in the garage.

**Figure 19 sensors-20-01211-f019:**
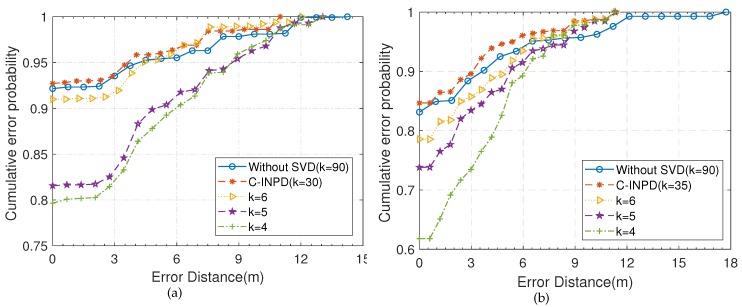
The positioning performance of different parameter K in SVD. (**a**) The CDF comparison in the integrated room. (**b**) The CDF comparison in the garage.

**Figure 20 sensors-20-01211-f020:**
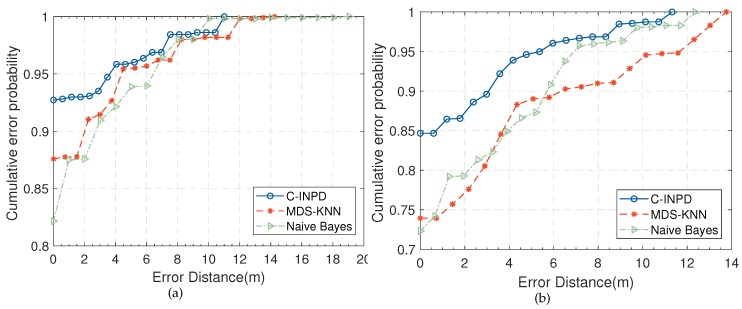
The performance comparision of different positioning systems. (**a**) The CDF comparison among C-InP, MDS-KNN and NB in the integrated room. (**b**) The CDF comparison among C-InP, MDS-KNN and NB in the garage.

**Table 1 sensors-20-01211-t001:** Main parameters of the scenarios.

Parameters	Spare Room	Integrated NLOS Room	Complex Garage
Size (m^2^)	7.2×5.8	16.5×8	23.4×12.5
Interval (m)	0.8	1.2	1.5
Access points	1	4	3
Cells	20	58	56

**Table 2 sensors-20-01211-t002:** Comparison of different preprocessing methods.

Scenes	Method	Accuracy	MDE (m)
Integrated NLOS room	C-InP	92.59%	0.49
	DBSCAN	90.60%	0.58
	Raw data	82.68%	0.88
Complex Garage	C-InP	84.60%	0.81
	DBSCAN	83.12%	0.92
	Raw data	79.30%	1.39

**Table 3 sensors-20-01211-t003:** Computational complexity of different positioning systems.

Scenes	System	Runtime (s)
Integrated room	C-InP	0.933
	Improved M-SVC	1.154
	MDS-KNN	0.691
	NB	0.550
Complex Garage	C-InP	0.798
	Improved M-SVC	0.979
	MDS-KNN	0.607
	NB	0.489

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
