# Peer review of "Indoor NLOS Positioning System Based on Enhanced CSI Feature with Intrusion Adaptability"

_sensors, 2020, doi:10.3390/s20041211_

Round 1

Reviewer 1 Report

The paper presents a very interesting work and in my opinion it is ready for publication, I only recommend that authors revise the document once there are some typos.

Author Response

Response to Reviewer 1 Comments

Point 1:  I recommend that authors revise the document once there are some typos.

Response 1: The several typos in the manuscript have been revised after rechecking the full text and they are highlighted on lines 41, 93, 157, 275, 301, 365 and 385, respectively.

We also would like to express our appreciation for taking the time to review carefully.

Reviewer 2 Report

The only remark I have to the authors of the article is an insufficient analysis of the results obtained by them. It would be good to have a separate Discussion section for this purpose, since the Conclusion section does not sufficiently describe all the benefits of their solution. It should be discussed how the use of the proposed solution can be seen in practice and what disadvantages it has.

Author Response

Response to Reviewer 2 Comments

Point 2: The only remark I have to the authors of the article is an insufficient analysis of the results obtained by them. It would be good to have a separate Discussion section for this purpose, since the Conclusion section does not sufficiently describe all the benefits of their solution. It should be discussed how the use of the proposed solution can be seen in practice and what disadvantages it has.

Response 2: We have adopted your rational suggestions, summarized and added a separate Discussion Section 4.4 (Discussion of the Proposed Methods in C-InP) to discuss the system performance and positioning accuracy of the optimization methods proposed in this paper in detail, and comprehensively considered the advantages and disadvantages of the system, as highlighted in line 445-470.

The several typos in the manuscript have been revised after rechecking the full text and they are highlighted on lines 41, 93, 211,157, 275, 301, 365 and 385, respectively.

We also would like to express our appreciation for taking the time to review carefully.

Reviewer 3 Report

The manuscript addresses the problem of exploiting channel state information (CSI) available from commercial WiFi devices for indoor localization and intrusion detection.
In particular, the authors propose a CSI-based indoor positioning system that performs intrusion detection in NLOS environment and subsequently positioning through fingerprinting.
Remarkably, the authors provided a comprehensive experimental validation in indoor environments like office and garage.

The manuscript is fairly well written and organized. The abstract and introduction are clear as well as the problem setup. The plots are nice, and the metrics used to describe the performance of the algorithm are correct.

Regarding the technical contribution, the idea proposed is new, although mixing some known tools, the final algorithm introduces sufficient novelties to worth publication. The performance obtained either for the detection as well as the localization accuracy, are remarkable.

The literature overview is fine and comprehensive.

I do not have any objections about the technical content as I was nota belt o spot any flaws.
In my opinion, the manuscript is suitable for publication.

Author Response

Response to Reviewer 3 Comments

Point 1: English language and style are fine/minor spell check required.

Response 1: The several typos in the manuscript have been revised after rechecking the full text and they are highlighted on lines 41, 93, 157, 211, 275, 301, 365 and 385, respectively.

We also would like to express our appreciation for taking the time to review carefully.